Vegetation Patterns and Competitive Dynamics along Elevation Gradients: Interactions between Environmental Factors and Vegetation in the Central Himalayas

3 4 5

1 2

Prashant Paudel<sup>1</sup>, Stefan Olin<sup>2</sup>, Mark Tjoelker<sup>1</sup>, Mikael Pontarp<sup>3</sup>, Daniel Metcalfe<sup>4</sup>, and Benjamin Smith<sup>1,2</sup>

<sup>1</sup> Hawkesbury Institute for the Environment, Western Sydney University, Penrith, NSW, Australia

<sup>2</sup> Department of Physical Geography and Ecosystem Science, Lund University, Lund, Sweden

<sup>3</sup>Department of Biology, Lund University, Lund, Sweden

<sup>4</sup> Department of Ecology and Environment Science, Umeå University

14 Corresponding Author: Benjamin Smith (ben.smith@nateko.lu.se)

15 16

24

26

35

#### 1. Abstract

Elevation gradients are generally characterized by a steady reduction in temperature with altitude, resulting in differences in growth conditions that often produce clear patterns of vegetation zonation. A south-north transect of the central Himalayas spans from a tropical climate in the south to alpine conditions in the north, offering an opportunity to investigate the relative roles of abiotic stress and competitive interactions in shaping plant community assembly. We hypothesise a shift from vegetation composition and productivity being characterised by realised niches, defined by competitive interactions at lower elevations, to physiological niches, shaped by stress (freezing temperature) at higher elevations. To investigate how these niche transitions influence community dynamics and ecosystem processes, we used a dynamic vegetation model with regional plant functional types (PFTs) parameterised with trait data, including allometric relationships. The model effectively captured spatial and temporal variability in vegetation structure and productivity along the gradient, with simulated patterns closely matching observed vegetation zonation across the transect. The establishment and performance of the PFTs were dependent on their climatic niche and the local abundance of competing PFTs, with persistence shaped by specific traits and adaptation strategies. At low elevations, where competitive interactions dominate, tropical shade-intolerant raingreen and tropical shade-tolerant evergreen PFTs dominated carbon mass production and vegetation cover. In contrast, shorter stature, evergreen phenology, and cold-tolerant types were favoured at high elevations, reflecting reduced interspecific

competition and physiological adaptation to low temperature stress. Along the elevation gradient, PFT functional diversity declined with elevation, but evenness in composition increased. Conversely, low-elevation communities supported higher functional diversity, yet vegetation structure and function (e.g., LAI, FPC, and carbon mass) were dominated by a few competitively superior PFTs. We conclude that vegetation dynamics along the temperature gradient are governed by a trade-off between competitive ability and stress tolerance, as reflected in shifts in structure, composition, and productivity, which are shaped by environmental conditions, functional traits, and adaptive strategies of the vegetation.

**Keywords**: Elevation gradient, temperature, competition, community composition, Himalayas, traits, dynamic vegetation model.

#### 2. Introduction

Elevation gradients are characterized by rapid changes in climatic conditions, particularly a decline in temperature with altitude (Zhu et al., 2022). With rising elevation, temperature changes predictably in accordance with the 'lapse rate', creating differences in growth conditions, which may be further exaggerated by small-scale variations in precipitation, topography, aspect, exposure, geology, soil types, and biogeochemical processes governing nutrient cycling and resource availability. These variations in growth conditions act as environmental filters (stress factors) (Asner et al., 2014; De Frenne et al., 2013; Zhu et al., 2022) which influence ecological processes and interactions that give rise to emergent eco-evolutionary patterns such as niche differentiation, functional trait distribution, plant strategies, and competitive exclusion (Asner et al., 2014; Muñoz Mazón et al., 2020). Distinct community composition and structure may result, with clear vegetation zonation shaped by the interplay of environmental filtering and competition interaction along the gradient.

Temperature gradients are among the most influential environmental axes shaping vegetation patterns globally. The Himalayas provide a striking example of this, with distinct zonation observed from the tropical lowlands in the south to the alpine conditions in the north. Cooler and wetter conditions at higher elevations, and hot with seasonally dry conditions at low elevations, play a significant role in shaping the

composition and survival of plant species (Måren et al., 2015). Additionally, competition for light, space, and soil resources drives vegetation function, diversity, and species composition along the elevation gradient (Maharjan et al., 2021; Thakur & Chawla, 2019), creating a biotic filter that results in different vegetation zones, community composition, and plant diversity (species richness and evenness). The combined effects of these abiotic and biotic filters on vegetation across elevations are reflected in differences in functional traits, growth strategies of vegetation, and community composition at different positions along the gradient (Midolo et al., 2019; Silva et al., 2023). As a result, vegetation at higher elevations often exhibits thicker leaves and frost tolerance adaptations such as reduced leaf area, increased epidermal thickness, and increased antifreeze proteins (Satyakam et al., 2022). In contrast, lower elevation species have larger leaves with higher specific leaf area (SLA), larger and wider branching patterns, and fast growth rates to take advantage of abundant resources and warm temperatures (Shah et al., 2019; Sigdel et al., 2022). Despite these clear vegetation patterns, the relative contributions of trait-mediated competitive interactions and adaptation to stress on vegetation structure, composition, and productivity remain poorly understood along the Himalayas' elevation gradient.

 The complex interaction between growth conditions and vegetation functional traits, and their role in structural dynamics and competitive interactions along this gradient, can be effectively captured by incorporating characteristic plant traits into dynamic vegetation models (DVMs) (De Paula et al., 2021; Sitch et al., 2003; Smith et al., 2014). By encoding the traits and response mechanisms of real-world vegetation for different plant functional types (PFTs) in these models and demonstrating agreement between modelled and observed vegetation structural and functional characteristics along the gradient, we may disentangle the environmental and community dynamic drivers of ecosystem productivity. DVMs represent vegetation as groups of functionally similar species, represented as PFTs, which share similar traits (morphological, ecophysiological, and life history characteristics) and climatic niches (bioclimatic limits for establishment or survival). Distinct traits and life history strategies, encoded as PFT parameters, influence their performance and interactions in model simulations, reflecting functional trade-offs in the adaptations of species that affect their performance when growing under different environmental conditions (Díaz et al., 2016; Pierce et al., 2013).

We tested how plant strategies reflected in their traits, competition, and interactions with environmental factors collectively shape vegetation structure, composition, and productivity along the Himalayas elevation gradient. We employed an individual-based dynamic vegetation model, LPJ-GUESS (Smith et al., 2001, 2014), to simulate structural, compositional, and functional variability along a south-north transect of the central Himalayas spanning from a tropical seasonal climate in the lowlands to alpine conditions. Our approach leverages empirical data on vegetation traits and life history strategies along the same gradient. The simulated PFTs were defined based on empirical data on vegetation traits and life history strategies for major species and taxa found along the Himalayan transect. By comparing simulated outcomes with field observations, we evaluate how well the mechanisms embedded in the model explain spatial patterns in structure, composition, and function variability. We also examine species richness and evenness as emergent properties of functional composition, reflecting how changing growth conditions influence diversity. Overall, this study illustrates the complex interplay between growth conditions, traits, and competitive dynamics, thereby enhancing our understanding of the ecological processes that shape plant communities along the elevation gradient of the central Himalayas.

## **3. Methods**

## **3.1 Study site**

The study focuses on a species-rich elevation gradient of the central Himalayas (Figure 1). Along the gradient, temperature decreases by approximately 6.5 °C per vertical kilometer, reflecting the standard lapse rate, where the average annual temperature in the tropical zone is 28 °C and around 10 °C in the alpine region (MoFSC, 2016; Poudel et al., 2020). Nearly 80% of the total annual rainfall occurs during the monsoon (June to September) (Maharjan et al., 2021), where average annual rainfall ranges from 165 mm (northern end of the gradient, i.e., Trans-Himalayan region) to 5244 mm (1550-2000 m altitude) in the lower and middle part of the gradient (Luitel et al., 2020; Poudel et al., 2020). Precipitation peaks at around 1000 m altitude asl and then rapidly decreases with an increase in elevation (Luitel et al., 2020; Poudel et al., 2020). Vegetation follows the temperature patterns ranging from tropical (24 °C) to temperate forests and to colder sub-alpine vegetation (6.9 °C) (Shrestha et al., 2015), where growth condition varies significantly due to variations in temperature and precipitation patterns. Within a horizontal span of 100 km along the gradient, 160 different tree species were recorded in

a plot-level survey conducted in forest areas, where vegetation composition and dominance changes with elevation (DFRS, 2015; Pokhrel and Sherpa, 2020).

Figure 1: Map of the study area with major global Biomes (data source: Olson et al., 2001), Forest types (data source: <a href="https://rds.icimod.org/">https://rds.icimod.org/</a>), trait data observed sites (Maharjan et al., 2021), and simulated grids along the elevation gradient of the central Himalayas.

## 3.2 Vegetation model description and customization

The Lund-Potsdam-Jena General Ecosystem Simulator (LPJ-GUESS) is a process-based dynamic vegetation model used for simulating and predicting ecosystem responses to environmental changes at the regional or global level based on local, neighbourhood-scale interactions among simulated plants (Smith et al., 2001, 2014). It represents generalized ecophysiological processes such as photosynthesis, autotrophic (plant) and heterotrophic (soil) respiration, carbon, water, and nitrogen cycling. The model adopts gap dynamics theory (Bugmann et al., 1996; Scherstjanoi et al., 2014) to simulate tree population dynamics emerging from the balance of plant establishment, growth, and mortality (De Paula et al., 2021; Sitch et al., 2003; Smith et al., 2001). The model is applied across a continuous geographic grid. Vegetation in each grid cell is represented as a

 mixture of PFTs whose potential distribution in climate space is governed by bioclimatic envelopes for the establishment and survival. Within its bioclimatic envelope, the presence and abundance of a PFT are additionally subject to vegetation dynamics determined by the interactions between co-occurring individuals of other PFTs and the cascading effects on carbon assimilation and allocation, reproduction, and survival of individuals co-occurring within replicate local patches, nominally 0.1 ha in size. The overall vegetation of a grid cell is aggregated across multiple patches (here 15), representing random samples of the wider landscape of the grid cell. PFT-specific parameters and growth strategies determine performance under different climates, CO<sub>2</sub> concentrations, and stages of vegetation development. The model accounts for structural responses to competitive and environmental conditions through adaptive allometric relations (DBH-Height, DBH-Crown Area, DBH-Crown Volume) and a dynamic bole height scheme for each cohort. This allows allometric scaling and carbon allocation to respond dynamically to the competition (crowding) conditions defined by the availability of photosynthetically active radiation (Paudel et al., Unpublished).

To represent the diversity of vegetation composition along the Himalayan elevation gradient, we modified the following features to customize the model for application to our study area.

- The default model has 12 PFTs, defined to represent the dominant biomes of the world. For this study, we defined a new set of regional tree PFTs tailored to local conditions using a multivariate clustering approach. These PFTs represented major hypothesised strategies and adaptation mechanisms employed by vegetation to cope with competition (biotic stress) and harsh climatic conditions (abiotic stress) along the elevation gradient. Tree PFT parameters and their derivation are further discussed in Section 3.3 below. Both C<sub>3</sub> and C<sub>4</sub> grass PFTs with default parameter values (defined for the global level) were retained for simulations (Peng et al., 2024).
- Bioclimatic limits (mean minimum and maximum temperature for the 20 years of
  coldest month for establishment and survival; mean minimum warmest month
  temperature for establishment) control each PFT's establishment and survival in
  a given grid cell. The model defines these limits for global biomes ranging from
  tropical to boreal ecosystems. For this study, four climatic limits are defined:

tropical, subtropical, temperate, and alpine. Climatic ranges for our study were adopted from Jackson (1994) and modified based on the ranges recorded by Maharjan et al. (2021).

## 3.3 Data Sources for model input and parameterisation

The CRU-JRA (v2.4.5d) global gridded climate dataset was downscaled to 3 km spatial resolution using bicubic interpolation (Latombe et al., 2018) and used as climate-forcing data for our model simulations. The CRU-JRA is a gridded daily dataset with  $0.5^{\circ} \times 0.5^{\circ}$  spatial resolution from 2001 to 2022 (Araghi and Martinez, 2024). We used monthly mean air temperature, precipitation, wind speed, incoming solar radiation, specific humidity, number of wet days, and minimum and maximum temperature as inputs. All variables except for precipitation were interpolated to daily values; for precipitation, the monthly sum was divided equally across the number of wet days per month. Soil properties and the atmospheric nitrogen deposition rates (Lamarque et al., 2013) were configured using  $0.5^{\circ} \times 0.5^{\circ}$  spatial resolution. Annual atmospheric CO<sub>2</sub> concentration data from NOAA (1901-022) are used as input data (Friedlingstein et al., 2023).

Trait values of the 31 most abundant tree species - based on the frequency of observation and total carbon contribution identified by Maharjan et al. (2021) were compiled from Maharjan et al. (2021), Jackson (1994), and Thakur & Phulara (2014). Tree allometry data (DBH, total tree height, crown radius, crown height) were compiled from the Tallo database (a global tree allometry and crown architecture database) (Jucker et al., 2022) and BADD (a biomass and allometry database for woody plants) (Falster et al., 2015). Elevation data across the simulated grid was extracted from Earth Resources Observation and Science (EROS) Center (2000) and was used for plotting simulated outputs.

A divisive hierarchical clustering approach was used to group tree species into distinct PFTs based on similarities in traits and life-history strategies. We clustered tree species into 13 functional groups based on their dominant temperature range (tropical, subtropical, temperate or alpine), leaf phenology (evergreen, raingreen, summergreen, broadleaved, and conifers), life history strategies (growing fast and slow, late and early successional species), shade requirement (shade tolerant, intermediate tolerant and intolerant); drought resistance class (drought tolerance, drought sensitive), as well as

traits such as wood density and total tree height (Supplementary Table 1). The resultant PFTs are designed to represent the functional diversity required to simulate structure, composition, and productivity along the elevation gradient.

The following parameters were updated for each tree PFT: leaf phenology, drought tolerance, wood density, SLA, shade tolerance, leaf longevity, and leaf turnover rate (Table 1). The values of these parameters, compiled from the sources mentioned above, were averaged across species included in the PFTs emerging from the clustering procedure described above. In the model, shade tolerance is linked to life history strategies, which influence growth, reproduction, and survival of PFTs across different light environments. Here, parameters and their values (Supplementary Table 2), as defined by Hickler et al. (2004), were adapted in model simulation to represent these dynamics. Similarly, quantile regression was used to estimate allometric scaling parameters (DBH relative to height, DBH relative to crown area, and DBH relative to crown volume) under three different stand crowding conditions (5%, 50% and 95%), allowing us to evaluate structural response to competition and their influence on ecosystem dynamics.

Table 1: Tree PFTs and parameter values used for simulation, including bioclimatic limits, Shade tolerance parameters and their values, and allometric relations (see Supplementary Table 1 & 2 for details).

| PFTs                                | Parameters      |           |            |                        |                     |                 |
|-------------------------------------|-----------------|-----------|------------|------------------------|---------------------|-----------------|
|                                     | Leaf            | Leaf      | Shade      | Wood                   | SLA                 | Leaf Turnover   |
|                                     | phenology       | longevity | tolerance  | density                | (m <sup>2</sup>     | (fraction/year) |
|                                     |                 | (years)   |            | (kgC m <sup>-3</sup> ) | kgC <sup>-1</sup> ) |                 |
| Alpine broadleaved evergreen (ABE)  | evergreen       | 2         | intolerant | 273                    | 13                  | 0.5             |
| Alpine needleleaved (ANE)           | evergreen       | 3         | tolerant   | 280                    | 9                   | 0.33            |
| Temperate<br>summergreen<br>(TeBSG) | summergr<br>een | 0.5       | intolerant | 265                    | 18                  | 1               |
| Temperate                           |                 | 2         | Tolerant   | 281                    | 15                  | 0.5             |
| broadleaved shade                   | evergreen       | 2         | Toleralit  | 201                    | 13                  | 0.5             |
| tolerant evergreen                  | CVCIGICCII      |           |            |                        |                     |                 |
| (TeBEt)                             |                 |           |            |                        |                     |                 |
| Temperate                           | evergreen       | 2         | intolerant | 282                    | 15                  | 0.5             |
| broadleaved shade                   |                 |           |            |                        |                     |                 |
| intolerant evergreen                |                 |           |            |                        |                     |                 |
| (TeIBE)                             |                 |           |            |                        |                     |                 |
| Temperate                           | evergreen       | 3         | intolerant | 235                    | 18                  | 0.33            |
| needleleaved                        |                 |           |            |                        |                     |                 |
| (TeNE)                              |                 |           |            |                        |                     |                 |

| PFTs                | Parameters |           |              |                        |         |                 |
|---------------------|------------|-----------|--------------|------------------------|---------|-----------------|
|                     | Leaf       | Leaf      | Shade        | Wood                   | SLA     | Leaf Turnover   |
|                     | phenology  | longevity | tolerance    | density                | $(m^2)$ | (fraction/year) |
|                     |            | (years)   |              | (kgC m <sup>-3</sup> ) | kgC-1)  |                 |
| Sub-tropical        | evergreen  | 3         | intolerant   | 210                    | 10      | 0.33            |
| needleleaved        |            |           |              |                        |         |                 |
| (STrNE)             |            |           |              |                        |         |                 |
| Sub-tropical rain   | raingreen  | 0.5       | intolerant   | 200                    | 25      | 1               |
| green (STrRG)       |            |           |              |                        |         |                 |
| Sub-tropical shade- | evergreen  | 1         | intermediate | 265                    | 20.85   | 1               |
| tolerant evergreen  |            |           |              |                        |         |                 |
| (STrIBE)            |            |           |              |                        |         |                 |
| Sub-tropical        | evergreen  | 2         | tolerant     | 145                    | 17      | 0.5             |
| broadleaved         |            |           |              |                        |         |                 |
| evergreen (STrBE)   |            |           |              |                        |         |                 |
| Tropical            | evergreen  | 2         | tolerant     | 290                    | 22      | 05              |
| broadleaved shade-  |            |           |              |                        |         |                 |
| tolerant (TrBE)     |            |           |              |                        |         |                 |
| Tropical            | summergr   | 0.5       | intolerant   | 247                    | 21      | 1               |
| Summergreen         | een        |           |              |                        |         |                 |
| (TrBSG)             |            |           |              |                        |         |                 |
| Tropical            | raingreen  | 1         | intolerant   | 245                    | 18      | 1               |
| broadleaved         |            |           |              |                        |         |                 |
| raingreen (TrBRG)   |            |           |              |                        |         |                 |

## 3.4 Simulation protocol and model validation

Using the aforementioned forcing data and PFTs parameterized with traits, the model was run with 15 patches in each grid cell of 1000 m², simulating the period from 1901 to 2022. A spin-up of 500 years, recycling the first 30 years of the observed climate data set, was performed to achieve an initial steady state for vegetation structure. We ran LPJ-GUESS in cohort mode (Smith et al., 2001, 2014), using the BLAZE fire model to account for the impacts of weather-related fire disturbances on vegetation structure (Rabin et al., 2017) and applied a generic return interval of 100 years for patch-destroying disturbances, following Pugh et al.(2019).

We implemented a neighbour removal experiment in the model to assess the effects of competitive neighbour individuals and PFTs on the performance of the selected PFTs (Monteux et al., 2024). Individuals of all other woody PFTs except the PFT of interest were removed from the simulation after model year 1950. C<sub>3</sub> grass or C<sub>4</sub> grass forming the understory of the woody stand was retained. However, to allow the ecosystem to reequilibrate and account for the effects caused by removing neighbours, we allowed target PFTs to grow for another 50 years (i.e., model year 2000) until the ecosystem recovered and productivity stabilized according to the prevailing environmental conditions.

265

269 270

271

272 273

274

275

276

277 278

We used Bi & Zhou (2022)'s gross primary productivity (GPP), produced using the leaf light use efficiency model from 2010 to 2020 (DRYAD), to compare with simulated GPP from 2010 to 2020 along the elevation gradient. Plot level above-ground carbon biomass data by Khanal & Boer (2023) estimated from the national forest inventory was compared with simulated patch-level carbon biomass from the model. Similarly, the bole height (height up to the first branch) measured and elevation range recorded by Maharjan et al. (2021) along the studied gradient were compared with simulated values to evaluate the model's ability to capture structure and compositional variability.

3.5 Competition index and rank abundance curve (RAC) of FPC for evenness

An index of competition was calculated for each PFT to quantify the effects of neighbour removal on the performance of the target PFT. The target PFT's performance was evaluated using simulated carbon mass production from 2000 to 2020 with and without competition. The competition index (CI) was calculated using the index matrices approach of Avolio et al. (2019) and Brooker & Kikvidze (2008) with modifications. We modified the equation by Avolio et al. (2019) and Brooker & Kikvidze (2008) to more intuitively represent each PFT's optimum competitive capacity relative to potentially cooccurring woody PFTs. A CI value of 1 indicates no effect from competitors, while a value close to 0 indicates a significant impact from the competitor's presence.

279 280

282

281 
$$CI_i = \left[1 - \frac{c_{mass-Ni} - c_{mass+Ni}}{max (c_{mass-Ni}, c_{mass+Ni})}\right] \dots (Equ i)$$

where CI is the competition index in year i, Cmass-N and Cmass+N are the carbon masses of the target PFTs in the presence (+N) and absence (-N) of a competitor in year i.

283 284 285

289

The rank abundance curve (RAC) of relative foliar projective cover (FPC) of each tree PFT present in a grid cell was used to quantify species evenness, defined as similarity in local abundances among PFTs along the elevation gradient. Following Avolio et al. (2019), Smith & Wilson (1996) and Whittaker (1965), the natural logarithm of relative FPC was plotted against the inverse of PFT rank, and a regression between rank and log FPC was used to calculate the slope of the curve in each grid point using simulated FPC from 2000 to 2022. The slope of the RAC was used as a robust (independent of species richness) measure of evenness by Smith & Wilson (1996). A steeper slope would indicate greater

dominance by one or a few PFTs relative to other co-occurring PFTs, while a flatter slope indicates greater evenness. Evenness was plotted against elevation to examine the patterns of change in evenness in relation to changes in growth conditions. Similarly, the number of PFTs in each latitude band was used as an indicator of functional richness, representing the diversity of ecological strategies. We hypothesise that with an increase in elevation, evenness increases due to reduced competition, and the abundance of PFTs decreases as environmental stress limits species adaptations in higher elevations.

### 4. Results

## 4.1 Spatial variability in productivity and structure along the gradient

Simulated annual GPP showed strong agreement with GPP estimated by Bi and Zhou et al. (2022) using the leaf light use efficiency model, both displaying a gradual increase in productivity up to approximately 2500 meters. In both datasets, at the northern end of the gradient, annual GPP decreased sharply as temperature declined, with some grids having very low productivity (GPP value close to zero) (Figure 2). These patterns align with shortening of productive seasons and decreasing temperature in alpine zones, reflecting a shift from productive lower elevations to physiological constraints (Supplementary Figures 2 & 3 for monthly GPP) at higher elevations.

Figure 2: Simulated annual GPP  $(0.03^\circ)$  and annual GPP from the DRYAD database  $(0.05^\circ)$  from 2010 to 2020 across grids along the elevation gradient of the Himalayas.

Simulated above-ground biomass at the patch level closely matched observed plot-level above-ground biomass in the forest area across the gradient, indicating that the model effectively captures spatial variability in carbon mass distribution (Figure 3). However, in the northern regions of the gradient, where *Rhododendron* PFTs dominate, the model underestimated the observed biomass (Figure 3). In contrast, the model slightly overestimated above-ground carbon biomass in the mid-latitude range  $(27.4 - 28.0 \, ^{\circ}\text{N})$ , corresponding to areas with higher PFT abundance. This mid-elevation peak in biomass accumulation, followed by a decline at higher elevation, suggests increasing physiological constraints under alpine conditions. Notably, the patch-level above-ground biomass did not show any patterns with the simulated age of the patch. In some simulated grids, young patches exhibited higher above-ground carbon productivity compared to older ones, suggesting that trait-based responses and environmental conditions have a stronger influence on productivity than patch age.

Figure 3: Simulated above-ground carbon mass per patch (2010-2015) with the age of the patch (in years) and observed above-ground carbon mass per plot in forest areas along the elevation gradient of the Himalayas.

Similar to productivity and biomass, vegetation structural attributes also varied along the elevation gradient. Simulated leaf area index (LAI) for tree PFTs increases gradually along the gradient from around 5 m $^2$  m $^{-2}$  to around 8 m $^2$  m $^{-2}$ , primarily due to the presence of evergreen PFTs at higher elevations, whose LAI remains constant throughout the year. Six PFTs contribute the most to LAI, each with varying zones (elevational range) of dominance. In the southern part of the gradient, two PFTs - tropical broadleaved raingreen and tropical broadleaved evergreen - exhibit higher PFT-specific LAI. Subtropical needle-leaved PFTs form maximum LAI in mid-latitude ( $\sim 3.2$  m $^2$  m $^{-2}$ ), followed by alpine evergreen broadleaved ( $\sim 4$  m $^2$  m $^{-2}$ ) in higher elevation. C4 and C3 grasses contribute significantly to LAI along the gradient, with C3 grasses being dominant in cold regions, with an LAI of 3.2 m $^2$  m $^{-2}$  (Figure 4).

Figure 4: Boxplot of Simulated LAI by PFTs and line plot of LAI (line represents mean LAI for each PFT and dots representing LAI at simulated grid across latitude) across the elevation gradient by PFTs.

The comparison of simulated (2010-2015) and measured bole heights shows that both follow the same patterns, although the model exhibits more pronounced variability across most PFTs (Figure 5). It shows that bole height varies with PFTs, with tropical broad-leaved raingreen (TrBRG) having a large bole height, followed by temperate shade-intolerant evergreen (TeIBE) (Figure 5). The overall mean bole height remains similar up to the temperate zone (approximately up to 3500 m), after which it decreases sharply, particularly in the upslope alpine zone, where bole heights remain smaller (Figure 5). This overall pattern of taller bole height at lower elevation likely reflects intensive competition for light, where individuals grow taller to avoid shading. In contrast, the decline in bole height in higher elevations is consistent with both physiological constraints and adaptive strategies that favour short bole height under alpine conditions.

Figure 5: Boxplot of simulated and observed bole height (median with ranges and outliers) and bole height for PFTs (dot denoting the bole height of each cohort of PFTs simulated in the grid), along with the mean bole height (red line) across PFTs along the elevation gradient.

## 4.3 PFT performance and competition index along the elevation gradient

The result indicates that the total carbon mass production depends on both the abundance of PFTs' and their location within the elevation range. It shows that total carbon mass production at the ecosystem level is highest in lower elevations (around 1000 m), where climatic conditions supported a balanced contribution to productivity from broadleaved evergreen, deciduous trees, and conifers (Supplementary Figure 4 for total carbon mass production along the elevation). In lower elevations, tropical broadleaved raingreen PFTs had the maximum carbon mass, whereas temperate shade-intolerant evergreen PFTs had the maximum carbon mass production in temperate regions. In colder areas, where only two tree PFTs were adaptive within their prescribed bioclimatic limits, alpine broadleaved evergreens (*Rhododendron* species) have the maximum contribution to carbon mass production. These findings highlight that PFTs' composition and their contribution to carbon mass production are driven by their climatic niche (elevational range) and adaptive capacity to existing climatic and competition conditions (Figure 6).

Figure 6: Carbon mass production by each PFT along the latitude (the inset figure shows the PFTs' distribution ranges recorded by Maharjan et al., (2021) in the elevation gradient).

The mean CI plotted against latitude shows that removing competitors had different levels of impact on PFT performance in carbon mass production (Figure 7). PFTs' responses without competition differ within and outside their dominant climatic range. In their dominant growth regions, TrBRG and ABE were least impacted by the presence of neighbours, with CI values over 0.9 (Figure 7). In the lower elevation range, the presence of temperate PFTs was more random, and their performance was not enhanced by neighbour removal. For example, the CI values of TeIBE, TeBSG, and TeBEt were higher (~0.9), suggesting similar performance with and without competitors (Figure 7). In general, subtropical PFTs highly benefited from competitor removal in the model (Figure 7). There were no clear patterns and associations of PFTs' CI with average annual temperature, as PFTs' performance depends on climatic niche and neighbours' presence (Figure 7).

Figure 7: Mean competitive index of PFTs and mean annual temperature (green line) with standard deviation of temperature in the last 30 years (green shaded area) along the elevation gradient.

## 4.4 Community composition and evenness along the elevation gradient

The simulated results show that PFT composition and dominance change with elevation and associated temperature, where the number of PFTs decreases with an increase in elevation. In lower elevations, three PFTs - tropical broadleaved raingreen, tropical broadleaved evergreen, and C4 grass - dominate despite over ten PFTs in that area. At the higher end of the gradient, alpine needle-leaved and alpine evergreen broadleaved form tree crown cover, where C3 grass dominance increases with elevation, reaching up to 70% in FPC. Sub-tropical needleleaved and temperate shade-intolerant evergreen PFTs dominate the mid-elevation range, showing variability in PFT composition and distribution along the gradient. In the mid-elevation gradient, an area dominated by sub-tropical and temperate PFTs, PFT composition changes more frequently than the gradient's other ends (Figure 8).

Figure 8: PFTs distribution and composition by fractional projective cover along the elevation gradient (inset figure shows the PFTs distribution ranges recorded by Maharjan et al., 2021 in the elevation gradient).

The RAC coefficient value, calculated for each simulated grid using the FPC value of trees, shows that PFT's evenness increases rapidly along the elevation gradient starting from mid-elevation (after 2000 m) (Figure 9). As latitude increases, PFT abundance decreases significantly, from a maximum of 10 PFTs in lower elevations to just 2 in higher elevations (Figure 9). In the high PFT abundance area, the competition was higher, reflected by the competitive dominance of a few PFTs (Supplementary figure 5). For example, tropical broadleaved raingreen, tropical broadleaved evergreen in lower elevation, sub-tropical conifers, and subtropical intermediate shade tolerant broadleaves in the mid-elevation range (Supplementary figure 5). These patterns suggest a symmetrical competition for light and nutrients at lower elevation despite richness in species composition. In contrast, regions with low PFT abundance exhibited higher evenness (coefficient value close to 0), indicating reduced competition and a stronger role of environmental filtering. Here, plant adaptation to temperature and soil nutrient limitation becomes the primary driver of vegetation structure and composition. A more detailed breakdown of PFT-specific FPC

distributions and dominance patterns along the elevation gradient is provided in Supplementary Figure 5.

Figure 9: PFTs evenness (RAC coefficient) across latitude with the number of PFTs (present) in each simulated grid.

 Deciduousness and shade tolerance are two dominant plant adaptation mechanisms of vegetation along the elevation gradient. The proportion of shade-tolerant species distribution decreases with elevation. Intermediate shade tolerance (*Schima wallichii*) has maximum FPC in the mid-elevation ranges (1200 - 2500 m), suggesting that these elevation ranges offer a balance in light and temperature conditions, favouring flexible growth strategies. The deciduous broadleaved species form dominant crown coverage in lower elevation, their contribution to FPC decreases with increased elevation, where cold temperature limit their performance. Evergreen conifer's contribution to FPC increases with an increase in elevation up to 4500 m, i.e., up to where *Pinus* species mostly grow (Figure 10). At the cold end of the gradient, the alpine broadleaved evergreen, especially *Rhododendron*, contributes the maximum in FPC. The dominance of evergreen trees at higher elevations reflects an adaptation strategy to cold climates, where year-round leaf

retention allows for rapid photosynthetic response to short favourable growing periods. Notably, around 1000 m elevation (Figure 10), both shade tolerance and deciduousness exhibit high evenness in PFT distribution, indicating a transitional zone where multiple plant adaptation strategies coexist due to overlapping ecological niches shaped by complex interaction between plant traits and environmental conditions.

Figure 10: Distribution and composition of PFTs adaptation strategies (shade tolerance and deciduousness) and grasses (line represents mean and bar represents standard deviation) along the elevation gradient.

# 5. Discussion

We evaluated the complex interaction between growth conditions, PFT abundance, productivity, and competitive interactions by simulating these assumed factors and their interdependent dynamics along the Himalayan elevation gradient. As expected, our model predicted shifts in ecosystem structure, composition, and productivity along the Himalayan elevational gradient, driven by shifting interactions between climatic conditions, especially temperature and nutrient availability, and plant adaptation strategies. Competition between co-occurring PFTs in crowded stands influenced the emergent composition and productivity, particularly at lower elevations, where warmer temperature and higher resource availability allowed species to occupy their realised niches. In contrast, colder and nutrient-limited conditions at higher elevations constrained community composition and structure to a few cold-tolerant species with short bole heights, operating within their physiological niches, resulting in more even but

less diverse stands. These patterns support our hypothesis that vegetation structures, composition, and productivity along the elevation gradient are structured by a shift from realised niches, defined by competitive interactions at lower elevations, to physiological niches, shaped by stress (freezing temperature) at higher elevations.

482 483 484

490

492

497

505

479

480

481

The model reliably reproduced key ecological patterns along the elevation gradient, with strong agreement between simulated outcomes and observations or independent reconstructed variables, supporting the plausibility of the results and the underlying mechanisms. Simulated GPP decreased with elevation, in agreement with the estimate from the leaf light use efficiency model by Bi & Zhou (2022), reflecting the dependency of GPP on temperature and growing-season length. This decline aligned with the distribution of cold-tolerant PFTs such as Rhododendron (ABE) and Coniferous species (ANE) found in higher elevations, consistent with the National Forest Inventory, which identifies that Rhododendron and Abies are the two most dominant species in higher elevations of the Himalayas (DFRS, 2015). Although the model does not represent detailed soil chemistry, the presence of these PFTs in the simulation aligns with their known adaptation mechanisms to high pH and low nutrients, characterized by a wide distribution range. Conifers and Rhododendron species are native to the Himalayan range and exhibit a wide physiological niche space that allows adaptation in high pH and low nutrient soil from temperate to alpine (sub-alpine) habitats (Thakur et al., 2024). The simulated results showed that the above-ground carbon biomass gradually increases before a significant decline with elevation (temperature) due to climatic stress, which is consistent with the patterns reported by Thakur et al. (2024). The higher estimation of observed values at the higher elevation could arise from variability in topographic factors that create favourable microclimatic conditions for the growth of *Rhododendron* and *Abies* forests, particularly in deep gorges and on southern aspects. The Himalayas region is characterized by a unique environmental condition where temperature and soil water availability vary on a short spatial scale (Thakur & Chawla, 2019; Tito et al., 2020) resulting in structural diversity and competitive dynamics, allowing certain PFTs to extend their realised niche beyond broader patterns along the gradient.

508509510

511

Vegetation structure along the elevation gradient was significantly impacted by the environmental conditions. In lower elevations, where climatic conditions are more

513

514515

516 517

524525

526

527

528

529

favourable and soil nutrients are abundant, PFTs tend to exhibit higher bole heights as a strategy to maximize light interception, avoid asymmetric competition for light, and reduce shading for competitors. Additionally, taller bole height of trees in these conditions may result from a trade-off, where some resources are allocated away from other parts, such as stem and branch development. Pokhrel and Sherpa (2020) also found that tree height, DBH, and above-ground biomass are significantly associated with elevation along the elevation gradient of the central Himalayas. With the increase in altitude, tree growth declines, and light competition also declines, where trees tend to have similar basal area and total tree height (Coomes and Allen, 2007). In contrast, LAI increased with an increase in elevation. In higher elevations, an increase in LAI was associated with the abundance of evergreen vegetation (Figure 5 & 10), especially Rhododendron and Abies species. Rhododendron presence in higher elevations is linked with survival strategies, including heat dissipation to avoid damage by excessive radiation in warm seasons and physiological mechanisms such as increasing intercellular fluid concentration and using reactive oxygen to withstand chilling temperatures (Li et al., 2022). However, these strategies are explicitly not represented in our model. Additionally, traits such as reduced height, smaller individual leaf area, lower SLA, and trade-offs in vessel diameter and density could also facilitate the wide distribution of Rhododendron species (Pandey et al., 2021).

530531532

The simulated vegetation community exhibited a distinct shift in compositional patterns across the elevation gradient. In lower elevations, the coexistence of a large number of PFTs resulted in higher functional diversity, likely due to favourable growth conditions, as the presence of large PFTs within their physiological niches led to higher competitive dominance by a few PFTs. Among 10 different PFTs, *Shorea robusta (PFT-TrBRG)* emerged as the dominant climax species at the lower elevations, showing consistency with national forest inventory data, which shows that *Shorea robusta* and its associated species, such as *Terminalia alata, Mollunthous philippines, and Lagerstroemina parviflora,* are the most common and productive in terms of biomass in the southern part of the central Himalayas (DFRS, 2015). Its competitive dominance and higher productivity were associated with drought and fire resistance (Gautam and Devoe, 2006). In contrast, higher elevations supported only a few PFTs (Supplementary Figure 5), with community composition shaped by abiotic stress such as low temperature and short growing seasons (Figure 7).

547

548 549

550

551 552

553554

555556

557

Ahmad et al. (2025) also stated that functional diversity is higher in lower elevation, even though species richness and phylogenetic diversity are higher in mid-elevation across the Himalayas. This pattern is broadly consistent with the stress-gradient hypothesis, which posits that abiotic stress dominates in harsher conditions, and competition is more influential in a benign environment (Bertness and Callaway, 1994). Additionally, the simulated PFTs composition is consistent with Mid-domain effects (Colwell and Lees, 2000; Smith and Wilson, 1996) with a large number of PFT peaks at the intermediate elevation range (1000- 2000 m), likely due to geometric constraints on range overlap (Figures 6, 8, and 10). The overall species composition pattern showed a monotonic decrease in species richness and diversity with increasing elevation gradient, with higher heterogeneity and the presence of unique species adapted to extreme climatic conditions (Sekar et al., 2024). At the lower end of the gradient, with more seasonal rainfall and dry conditions, the vegetation was characterized by lower evenness and higher competitive dominance, as PFTs relied on realised niche occupation (Figure 8).

558559560

561

562 563

564 565

The PFTs' performance and dominance along the gradient depended on the climatic condition and their allometric relations. Overall, tropical broadleaved raingreen (TrBRG) was the most productive, followed by temperate evergreen (TeIBE) and alpine evergreen (ABE) across the elevation gradient. This result is consistent with the national forest inventory report, which states that Shorea robusta (37.83 ton ha-1), Quercus species (46.09 ton ha-1), and Rhododendron species (11.22 ton ha-1) are the most productive species in Nepal, with significant contributions from Pinus species (DFRS, 2015). Khanal et al. (2024) also concluded that species growth and carbon mass production depends on the wood density and size-density relation. Rhododendron (ABE) demonstrated wide niche space, and Shorea robusta (TrBRG), with its strong climatic niche and role climax vegetation, illustrates how PFT performance is shaped by the climatic niche at the elevations in which the species occurs. The results show that competitive dominance was not visible in higher elevations, with no individual PFTs becoming dominant. Similar to our finding, Naud et al. (2019) also highlighted that no individual species becomes dominant with increased altitude when species richness decreases, which is similar in the Himalayas.

A range of mechanisms, including deciduousness, shade tolerance, and drought resistance, contributed to the presence and abundance of PFTS under different environmental conditions along the gradient in our simulations. In lower elevations, deciduousness as an adaptation to escape seasonal drought was dominant, whereas, in higher elevations, cold and shade tolerant PFTs were abundant, reflecting their functional strategy to adapt to stresses such as cooler temperatures, shorter growing seasons, and harsher growing conditions. Similarly, allometric traits such as crown dimension-DBH, height-DBH, and wood density affected competitive interaction and competitive dynamics. In lower elevations, PFT coverage in the crown showed a clear hierarchy with the differences in competitive dominance in productivity and FPC. Previous studies have shown that traits shape the distribution of vegetation (Maharajan et al., 2021), and competition determines productivity under favourable conditions (Sauter et al., 2021). Our study further emphasized that growth conditions, coupled with biotic and abiotic interactions, and trade-offs between growth and adaptation to multiple stresses drive the overall ecosystem functioning along the elevation gradient. This is reflected in differences in productivity and vegetation structure along the elevation gradient, with variation in species abundance and evenness according to environmental conditions.

**5.1 Limitations** 

We used high-resolution climate data (3 km) as forcing data to capture heterogeneity in climatic conditions along the elevation gradient. This approach successfully captured broad elevational patterns in the dynamics and composition of the ecosystem. However, the model does not fully capture the variability in microclimatic conditions created by small-scale topography. This may partially account for the underestimation of aboveground carbon stock, especially in high elevations. With the elevation increase, climatic heterogeneity amplifies with more diverse climatic conditions (Guan et al., 2024). Integrating slope and aspect in the model could enhance its ability to characterize vegetation dynamics and composition dynamics, particularly in mountain regions where variation in radiation, soil moisture, and temperature across slope orientations plays a crucial role.

Even though dynamic vegetation models like LPJ-GUESS integrate competition for space, light, and soil resources among neighbouring plants, this competitive framework may not

614

628

630

632

641

fully account for how trait-based plasticity alters competitive interactions under different growth conditions. This can limit the model's ability to capture the full range of ecosystem dynamics. Integrating the allometric relation defined based on growth (competition) conditions helped to better characterise above-ground competition for light and space, whereas competition for below-ground resources, i.e. water and nitrogen in the model, largely follows proportionately with plant size. In reality, differential root profiles, including deep water access via tap roots characteristic of certain tree taxa, and other factors such as mycorrhizal associations or release of root exudates to promote nutrient mineralisation and uptake are known to predict plant success in environments characterised by below-ground resource limitations (Freschet et al., 2021). Dynamic root allocation based on resource availability in different layers is not currently simulated in LPJ-GUESS. Integrating root trait data, especially the distribution of fine roots in different soil profiles, root and mycorrhizal association (De Paula et al., 2021), and interlinking them with soil depth, may better capture below-ground competition processes. The current version of LPJ-GUESS incorporates a climate-driven prognostic wildfire scheme (SIMFIRE-BLAZE; Rabin et al., 2017) which impacts the composition, structure, and dynamics of the vegetation. However, other forms of disturbance, such as wood cutting and managed fires, herbivory, insect pest impacts, and wind-throw shape vegetation composition and demography, compounding with the biophysical and ecological mechanisms included in the model (Brewer, 2011; Grime, 1973; Hall et al., 2012; Laurent et al., 2017).

6. Conclusions

After incorporating trait data and allometric relations reflecting regional PFTs from our Himalayan study region, we successfully reproduced spatial and temporal patterns in vegetation composition, structure, and productivity along the elevation gradient. Our model shows that environmental conditions, biotic and abiotic interactions, allometric relations, and associated functional trade-offs jointly shape ecosystem processes and drive the competition patterns and adaptation mechanisms of vegetation along the elevation gradient. At the productive end of the gradient, competitive interactions among woody PFTs in crowded stands had a strong influence on PFT performance and abundance. These interactions, together with the realised niches of PFTs, led to reduced evenness in PFT distribution, as certain PFTs become dominant in different climatic

645

646

647648

657

conditions. This did not translate into higher PFT richness in the way predicted by classical niche theory, suggesting favourable environmental conditions may buffer competitive exclusion and promote species coexistence despite competitive dominance hierarchies. In contrast, under more stressful conditions, only a few PFTs survive and grow, exhibiting higher evenness in composition and shorter stature. The increase in evenness with elevation reflects reduced crowding and weak asymmetric competition, with the surviving PFTs adapting and persisting by occupying their physiological niches in response to prevailing abiotic stress, particularly cold and freezing temperatures. Our findings highlight how variations in climatic conditions, resource availability, the climatic niche of PFTs, and unique adaptation mechanisms interact with PFT traits and adaptation strategies to shape vegetation composition patterns and productivity, thereby acting as overall controls on ecosystem function along the gradient. Further study, integrating heterogeneity in topographic conditions with different disturbances, along with a representation of underground competition (especially root profiles and groundwater dynamics), could enhance our understanding of ecosystem responses to global changes and extreme events, as well as their adaptation mechanisms in the central Himalayas.

660

664

## **Code and Data Availability**

The customized LPJ-GUESS version used in this study has been archived in the **LPJ-GUESS Zenodo community** [https://doi.org/10.5281/zenodo.17214801]. The forcing data, simulated output, that reproduce the analyses presented in the manuscript have been deposited in **Zenodo** [https://doi.org/10.5281/zenodo.17214851].

669

665

### **Author Contribution**

**PP:** conceptualization and design (lead); data curation (lead); simulation (lead); formal analysis (lead); writing – original draft (lead); writing – review and editing (lead). **SO:** Supervision (supporting); writing – review and editing (supporting). **MT:** Supervision (supporting); writing – review and editing (supporting); Supervision (supporting); writing – review and editing (supporting). **MP:** Supervision (supporting); writing – review and editing (supporting). **DM:** Supervision (supporting); writing – review and

675 writing – original draft (supporting); writing – review and editing (equal). 676 677 **Competing Interest** 678 The contact author has declared that none of the authors has any competing interests. 679 680 Financial support 681 This research has been supported by Western Sydney University as a PhD scholarship. 682 Stefan Olin was supported by the Modelling the Regional and Global Earth System 683 (MERGE). 684 7. Reference 685 686 Ahmad, M., Luo, Y.-H., Rathee, S., Spicer, R. A., Zhang, J., Wambulwa, M. C., Zhu, G.-F., 687 Cadotte, M. W., Wu, Z.-Y., Khan, S. M., Maity, D., Li, D.-Z., and Liu, J.: Multifaceted 688 plant diversity patterns across the Himalaya: Status and outlook, Plant Divers., 689 47, 529–543, https://doi.org/10.1016/j.pld.2025.04.003, 2025. 690 Araghi, A. and Martinez, C. J.: Evaluation of CRU-JRA gridded meteorological dataset for 691 modeling of wheat production systems in Iran, Int. J. Biometeorol., 68, 1201-692 1211, https://doi.org/10.1007/s00484-024-02659-9, 2024. 693 Asner, G. P., Anderson, C. B., Martin, R. E., Knapp, D. E., Tupayachi, R., Sinca, F., and Malhi, Y.: Landscape-scale changes in forest structure and functional traits along an 694 Andes-to-Amazon elevation gradient, Biogeosciences, 11, 843–856. 695 696 https://doi.org/10.5194/bg-11-843-2014, 2014. 697 Avolio, M. L., Forrestel, E. J., Chang, C. C., La Pierre, K. J., Burghardt, K. T., and Smith, M. D.: Demystifying dominant species, New Phytol., 223, 1106–1126, 698 https://doi.org/10.1111/nph.15789, 2019. 699 Bertness, M. D. and Callaway, R.: Positive interactions in communities, Trends Ecol. Evol., 700 701 9, 191–193, https://doi.org/10.1016/0169-5347(94)90088-4, 1994. Bi, W. and Zhou, Y.: A global 0.05° dataset for gross primary production of sunlit and 702 703 shaded vegetation canopies (1992–2020) (18), 704 https://doi.org/10.5061/DRYAD.DFN2Z352K, 2022. 705 Brewer, J. S.: Disturbance-mediated competition between perennial plants along a 706 resource supply gradient: Disturbances and competition, J. Ecol., 99, 1219–1228, 707 https://doi.org/10.1111/j.1365-2745.2011.01846.x, 2011. 708 Brooker, R. W. and Kikvidze, Z.: Importance: an overlooked concept in plant interaction 709 research, J. Ecol., 96, 703-708, https://doi.org/10.1111/j.1365-710 2745.2008.01373.x, 2008.

editing (supporting). BS: Supervision (lead); conceptualization and design (supporting);

720

721

722 723

737

738

739

740

- Bugmann, H., Fischlin, A., and Kienast, F.: Model convergence and state variable update in forest gap models, Ecol. Model., 89, 197–208, https://doi.org/10.1016/0304-3800(95)00135-2, 1996.
- Colwell, Robert. K. and Lees, D. C.: The mid-domain effect: geometric constraints on the geography of species richness, Trends Ecol. Evol., 15, 70–76, https://doi.org/10.1016/S0169-5347(99)01767-X, 2000.
- Coomes, D. A. and Allen, R. B.: Effects of size, competition and altitude on tree growth, J. 718 Ecol., 95, 1084–1097, https://doi.org/10.1111/j.1365-2745.2007.01280.x, 2007.
  - De Frenne, P., Graae, B. J., Rodríguez-Sánchez, F., Kolb, A., Chabrerie, O., Decocq, G., De Kort, H., De Schrijver, A., Diekmann, M., Eriksson, O., Gruwez, R., Hermy, M., Lenoir, J., Plue, J., Coomes, D. A., and Verheyen, K.: Latitudinal gradients as natural laboratories to infer species' responses to temperature, J. Ecol., 101, 784–795, https://doi.org/10.1111/1365-2745.12074, 2013.
- De Paula, M. D., Forrest, M., Langan, L., Bendix, J., Homeier, J., Velescu, A., Wilcke, W., and Hickler, T.: Nutrient cycling drives plant community trait assembly and ecosystem functioning in a tropical mountain biodiversity hotspot, New Phytol., 232, 551– 566, https://doi.org/10.1111/nph.17600, 2021.
- DFRS: STATE of NEPAL'S FORESTS, Department of Forest Research and Survey, Kathmandu, https://doi.org/978-9937-8896-3-6, 2015.
- Díaz, S., Kattge, J., Cornelissen, J. H. C., Wright, I. J., Lavorel, S., Dray, S., Reu, B., Kleyer, M.,
  Wirth, C., Colin Prentice, I., Garnier, E., Bönisch, G., Westoby, M., Poorter, H., Reich,
  P. B., Moles, A. T., Dickie, J., Gillison, A. N., Zanne, A. E., Chave, J., Joseph Wright, S.,
  Sheremet'ev, S. N., Jactel, H., Baraloto, C., Cerabolini, B., Pierce, S., Shipley, B.,
  Kirkup, D., Casanoves, F., Joswig, J. S., Günther, A., Falczuk, V., Rüger, N., Mahecha,
  M. D., and Gorné, L. D.: The global spectrum of plant form and function, Nature,
  529, 167–171, https://doi.org/10.1038/nature16489, 2016.
  - Earth Resources Observation and Science (EROS) Center: Shuttle Radar Topography Mission (SRTM) 1 Arc-Second Global, https://doi.org/10.5066/F7PR7TFT, 2000. Falster, D. S., Duursma, R. A., Ishihara, M. I., Barneche, D. R., FitzJohn, R. G., Vårhammar,

A., Aiba, M., Ando, M., Anten, N., Aspinwall, M. J., Baltzer, J. L., Baraloto, C.,

- Battaglia, M., Battles, J. J., Bond-Lamberty, B., Van Breugel, M., Camac, J., Claveau,
  Y., Coll, L., Dannoura, M., Delagrange, S., Domec, J.-C., Fatemi, F., Feng, W.,
  Gargaglione, V., Goto, Y., Hagihara, A., Hall, J. S., Hamilton, S., Harja, D., Hiura, T.,
  Holdaway, R., Hutley, L. S., Ichie, T., Jokela, E. J., Kantola, A., Kelly, J. W. G., Kenzo, T.,
  King, D., Kloeppel, B. D., Kohyama, T., Komiyama, A., Laclau, J.-P., Lusk, C. H.,
  Maguire, D. A., Le Maire, G., Mäkelä, A., Markesteijn, L., Marshall, J., McCulloh, K.,
- Miyata, I., Mokany, K., Mori, S., Myster, R. W., Nagano, M., Naidu, S. L., Nouvellon, Y., O'Grady, A. P., O'Hara, K. L., Ohtsuka, T., Osada, N., Osunkoya, O. O., Peri, P. L.,
- Petritan, A. M., Poorter, L., Portsmuth, A., Potvin, C., Ransijn, J., Reid, D., Ribeiro, S.
- C., Roberts, S. D., Rodríguez, R., Saldaña-Acosta, A., Santa-Regina, I., Sasa, K.,
- Selaya, N. G., Sillett, S. C., Sterck, F., Takagi, K., Tange, T., Tanouchi, H., Tissue, D.,
  T52 Umehara, T., Utsugi, H., Vadeboncoeur, M. A., Valladares, F., Vanninen, P., Wang, J.
- R., Wenk, E., Williams, R., De Aquino Ximenes, F., Yamaba, A., Yamada, T., 754 Yamakura, T., Yanai, R. D., and York, R. A.: BAAD: a Biomass And Allometry
- Database for woody plants: Ecological Archives E096-128, Ecology, 96, 1445– 756 1445, https://doi.org/10.1890/14-1889.1, 2015.
- Freschet, G. T., Roumet, C., Comas, L. H., Weemstra, M., Bengough, A. G., Rewald, B.,
  Bardgett, R. D., De Deyn, G. B., Johnson, D., Klimešová, J., Lukac, M., McCormack, M.
  L., Meier, I. C., Pagès, L., Poorter, H., Prieto, I., Wurzburger, N., Zadworny, M.,

```
               Bagniewska-Zadworna, A., Blancaflor, E. B., Brunner, I., Gessler, A., Hobbie, S. E.,
               Iversen, C. M., Mommer, L., Picon-Cochard, C., Postma, J. A., Rose, L., Ryser, P.,
               Scherer-Lorenzen, M., Soudzilovskaia, N. A., Sun, T., Valverde-Barrantes, O. J.,
               Weigelt, A., York, L. M., and Stokes, A.: Root traits as drivers of plant and
               ecosystem functioning: current understanding, pitfalls and future research needs,
               New Phytol., 232, 1123–1158, https://doi.org/10.1111/nph.17072, 2021.
       Friedlingstein, P., O'Sullivan, M., Jones, M. W., Andrew, R. M., Bakker, D. C. E., Hauck, J.,
               Landschützer, P., Le Quéré, C., Luijkx, I. T., Peters, G. P., Peters, W., Pongratz, J.,
               Schwingshackl, C., Sitch, S., Canadell, J. G., Ciais, P., Jackson, R. B., Alin, S. R.,
               Anthoni, P., Barbero, L., Bates, N. R., Becker, M., Bellouin, N., Decharme, B., Bopp,
               L., Brasika, I. B. M., Cadule, P., Chamberlain, M. A., Chandra, N., Chau, T.-T.-T.,
               Chevallier, F., Chini, L. P., Cronin, M., Dou, X., Enyo, K., Evans, W., Falk, S., Feely, R.
               A., Feng, L., Ford, D. J., Gasser, T., Ghattas, J., Gkritzalis, T., Grassi, G., Gregor, L.,
               Gruber, N., Gürses, Ö., Harris, I., Hefner, M., Heinke, J., Houghton, R. A., Hurtt, G. C.,
               Iida, Y., Ilyina, T., Jacobson, A. R., Jain, A., Jarníková, T., Jersild, A., Jiang, F., Jin, Z.,
               Joos, F., Kato, E., Keeling, R. F., Kennedy, D., Klein Goldewijk, K., Knauer, J.,
               Korsbakken, J. I., Körtzinger, A., Lan, X., Lefèvre, N., Li, H., Liu, J., Liu, Z., Ma, L.,
               Marland, G., Mayot, N., McGuire, P. C., McKinley, G. A., Meyer, G., Morgan, E. J.,
               Munro, D. R., Nakaoka, S.-I., Niwa, Y., O'Brien, K. M., Olsen, A., Omar, A. M., Ono, T.,
               Paulsen, M., Pierrot, D., Pocock, K., Poulter, B., Powis, C. M., Rehder, G., Resplandy,
               L., Robertson, E., Rödenbeck, C., Rosan, T. M., Schwinger, J., Séférian, R., et al.:
               Global Carbon Budget 2023, Earth Syst. Sci. Data, 15, 5301–5369,
               https://doi.org/10.5194/essd-15-5301-2023, 2023.
       Gautam, K. H. and Devoe, N. N.: Ecological and anthropogenic niches of sal (Shorea
               robusta Gaertn. f.) forest and prospects for multiple-product forest management
               - a review, For. Int. J. For. Res., 79, 81-101,
               https://doi.org/10.1093/forestry/cpi063, 2006.
       Grime, J. P.: Competitive Exclusion in Herbaceous Vegetation, Nature, 242, 344-347,
               https://doi.org/10.1038/242344a0, 1973.
       Guan, Y., Liu, J., Cui, W., Chen, D., Zhang, J., Lu, H., Maeda, E. E., Zeng, Z., and Beck, H. E.:
               Elevation Regulates the Response of Climate Heterogeneity to Climate Change,
               Geophys. Res. Lett., 51, e2024GL109483,
               https://doi.org/10.1029/2024GL109483, 2024.
       Hall, A. R., Miller, A. D., Leggett, H. C., Roxburgh, S. H., Buckling, A., and Shea, K.:
               Diversity-disturbance relationships: frequency and intensity interact, Biol. Lett.,
               8, 768-771, https://doi.org/10.1098/rsbl.2012.0282, 2012.
       Hickler, T., Smith, B., Sykes, M. T., Davis, M. B., Sugita, S., and Walker, K.: USING A
               GENERALIZED VEGETATION MODEL TO SIMULATE VEGETATION DYNAMICS IN
               NORTHEASTERN USA, Ecology, 85, 519-530, https://doi.org/10.1890/02-0344,
               2004.
       Jackson, J. K.: Manual of Afforestation in Nepal, Second., Forest research and Survey
               Centre, Kathmandu, Nepal, 1994.
       Jucker, T., Fischer, F. J., Chave, J., Coomes, D. A., Caspersen, J., Ali, A., Loubota Panzou, G. J.,
               Feldpausch, T. R., Falster, D., Usoltsev, V. A., Adu-Bredu, S., Alves, L. F., Aminpour,
               M., Angoboy, I. B., Anten, N. P. R., Antin, C., Askari, Y., Muñoz, R., Ayyappan, N.,
               Balvanera, P., Banin, L., Barbier, N., Battles, J. J., Beeckman, H., Bocko, Y. E., Bond-
               Lamberty, B., Bongers, F., Bowers, S., Brade, T., van Breugel, M., Chantrain, A.,
               Chaudhary, R., Dai, J., Dalponte, M., Dimobe, K., Domec, J. C., Doucet, J. L.,
               Duursma, R. A., Enríquez, M., van Ewijk, K. Y., Farfán-Rios, W., Fayolle, A., Forni, E.,
```

| 809 | Forrester, D. I., Gilani, H., Godlee, J. L., Gourlet-Fleury, S., Haeni, M., Hall, J. S., He, J.   |
|-----|---------------------------------------------------------------------------------------------------|
| 810 | K., Hemp, A., Hernández-Stefanoni, J. L., Higgins, S. I., Holdaway, R. J., Hussain, K.,           |
| 811 | Hutley, L. B., Ichie, T., Iida, Y., Jiang, H. sheng, Joshi, P. R., Kaboli, H., Larsary, M. K.,    |
| 812 | Kenzo, T., Kloeppel, B. D., Kohyama, T., Kunwar, S., Kuyah, S., Kvasnica, J., Lin, S.,            |
| 813 | Lines, E. R., Liu, H., Lorimer, C., Loumeto, J. J., Malhi, Y., Marshall, P. L., Mattsson, E.,     |
| 814 | Matula, R., Meave, J. A., Mensah, S., Mi, X., Momo, S., Moncrieff, G. R., Mora, F.,               |
| 815 | Nissanka, S. P., O'Hara, K. L., Pearce, S., Pelissier, R., Peri, P. L., Ploton, P., Poorter, L.,  |
| 816 | Pour, M. J., Pourbabaei, H., Dupuy-Rada, J. M., Ribeiro, S. C., Ryan, C., Sanaei, A.,             |
| 817 | Sanger, J., Schlund, M., Sellan, G., et al.: Tallo: A global tree allometry and crown             |
| 818 | architecture database, Glob. Change Biol., 28, 5254–5268,                                         |
| 819 | https://doi.org/10.1111/gcb.16302, 2022.                                                          |
| 820 | Khanal, S. and Boer, M. M.: Plot-level estimates of aboveground biomass and soil organic          |
| 821 | carbon stocks from Nepal's forest inventory, Sci. Data, 10, 406,                                  |
| 822 | https://doi.org/10.1038/s41597-023-02314-9, 2023.                                                 |
|     |                                                                                                   |
| 823 | Khanal, S., Nolan, R. H., Medlyn, B. E., and Boer, M. M.: Disentangling contributions of          |
| 824 | allometry, species composition and structure to high aboveground biomass                          |
| 825 | density of high-elevation forests, For. Ecol. Manag., 554, 121679,                                |
| 826 | https://doi.org/10.1016/j.foreco.2023.121679, 2024.                                               |
| 827 | Lamarque, JF., Shindell, D. T., Josse, B., Young, P. J., Cionni, I., Eyring, V., Bergmann, D.,    |
| 828 | Cameron-Smith, P., Collins, W. J., Doherty, R., Dalsoren, S., Faluvegi, G., Folberth, G.,         |
| 829 | Ghan, S. J., Horowitz, L. W., Lee, Y. H., MacKenzie, I. A., Nagashima, T., Naik, V.,              |
| 830 | Plummer, D., Righi, M., Rumbold, S. T., Schulz, M., Skeie, R. B., Stevenson, D. S.,               |
| 831 | Strode, S., Sudo, K., Szopa, S., Voulgarakis, A., and Zeng, G.: The Atmospheric                   |
| 832 | Chemistry and Climate Model Intercomparison Project (ACCMIP): overview and                        |
| 833 | description of models, simulations and climate diagnostics, Geosci. Model Dev., 6,                |
| 834 | 179–206, https://doi.org/10.5194/gmd-6-179-2013, 2013.                                            |
| 835 | Latombe, G., Burke, A., Vrac, M., Levavasseur, G., Dumas, C., Kageyama, M., and Ramstein,         |
| 836 | G.: Comparison of spatial downscaling methods of general circulation model                        |
| 837 | results to study climate variability during the Last Glacial Maximum, Geosci.                     |
| 838 | Model Dev., 11, 2563–2579, https://doi.org/10.5194/gmd-11-2563-2018, 2018.                        |
| 839 | Laurent, L., Mårell, A., Korboulewsky, N., Saïd, S., and Balandier, P.: How does disturbance      |
| 840 | affect the intensity and importance of plant competition along resource                           |
| 841 | gradients?, For. Ecol. Manag., 391, 239–245,                                                      |
| 842 | https://doi.org/10.1016/j.foreco.2017.02.003, 2017.                                               |
| 843 | Li, H., Guo, Q., Yang, L., Quan, H., and Wang, S.: Seasonal Eco-Physiology Characteristics of     |
| 844 | Four Evergreen Rhododendron Species to the Subalpine Habitats, Forests, 13,                       |
| 845 | 653, https://doi.org/10.3390/f13050653, 2022.                                                     |
| 846 | Luitel, D. R., Jha, P. K., Siwakoti, M., Shrestha, M. L., and Munniappan, R.: Climatic Trends     |
| 847 | in Di ff erent Bioclimatic Zones in the, Climate, 8, 1–18, 2020.                                  |
|     |                                                                                                   |
| 848 | Maharjan, S. K., Sterck, F. J., Dhakal, B. P., Makri, M., and Poorter, L.: Functional traits      |
| 849 | shape tree species distribution in the Himalayas, J. Ecol., 109, 3818–3834,                       |
| 850 | https://doi.org/10.1111/1365-2745.13759, 2021.                                                    |
| 851 | Måren, I. E., Karki, S., Prajapati, C., Yadav, R. K., and Shrestha, B. B.: Facing north or south: |
| 852 | Does slope aspect impact forest stand characteristics and soil properties in a                    |
| 853 | semiarid trans-Himalayan valley?, J. Arid Environ., 121, 112–123,                                 |
| 854 | https://doi.org/10.1016/j.jaridenv.2015.06.004, 2015.                                             |
| 855 | Midolo, G., De Frenne, P., Hölzel, N., and Wellstein, C.: Global patterns of intraspecific leaf   |
| 856 | trait responses to elevation, Glob. Change Biol., 25, 2485–2498,                                  |
| 857 | https://doi.org/10.1111/gcb.14646, 2019.                                                          |
|     |                                                                                                   |

889

- MoFSC: Conservation Landscapes of Nepal, Ministry of Forests and Soil Conservation, Kathmandu, Nepal, 2016.
- Monteux, S., Blume-Werry, G., Gavazov, K., Kirchhoff, L., Krab, E. J., Lett, S., Pedersen, E. P., and Väisänen, M.: Controlling biases in targeted plant removal experiments, New Phytol., 242, 1835–1845, https://doi.org/10.1111/nph.19386, 2024.
  - Muñoz Mazón, M., Klanderud, K., Finegan, B., Veintimilla, D., Bermeo, D., Murrieta, E., Delgado, D., and Sheil, D.: How forest structure varies with elevation in old growth and secondary forest in Costa Rica, For. Ecol. Manag., 469, 118191, https://doi.org/10.1016/j.foreco.2020.118191, 2020.
- Naud, L., Måsviken, J., Freire, S., Angerbjörn, A., Dalén, L., and Dalerum, F.: Altitude effects on spatial components of vascular plant diversity in a subarctic mountain tundra, Ecol. Evol., 9, 4783–4795, https://doi.org/10.1002/ece3.5081, 2019.
- Olson, D. M., Dinerstein, E., Wikramanayake, E. D., Burgess, N. D., Powell, G. V. N.,
  Underwood, E. C., D'amico, J. A., Itoua, I., Strand, H. E., Morrison, J. C., Loucks, C. J.,
  Allnutt, T. F., Ricketts, T. H., Kura, Y., Lamoreux, J. F., Wettengel, W. W., Hedao, P.,
  and Kassem, K. R.: Terrestrial Ecoregions of the World: A New Map of Life on
  Earth, BioScience, 51, 933, https://doi.org/10.1641/00063568(2001)051[0933:TEOTWA]2.0.CO;2, 2001.
- Pandey, M., Pathak, M. L., and Shrestha, B. B.: Morphological and wood anatomical traits
   of *Rhododendron lepidotum* Wall ex G. Don along the elevation gradients in Nepal
   Himalayas, Arct. Antarct. Alp. Res., 53, 35–47,
   https://doi.org/10.1080/15230430.2020.1859719, 2021.
- Paudel, P., Olin, S., Tjoelker, M., Pontarp, M., Metcalfe, D., and Smith, B.: Impacts of tree
   allometry on structure, composition, functioning and competitive interactions in
   savanna ecosystems on the Northern Australian Tropical Transect, J. Biogeogr.,
   Unpublished, Unpublished.
- Peng, S., Terrer, C., Smith, B., Ciais, P., Han, Q., Nan, J., Fisher, J. B., Chen, L., Deng, L., and Yu, K.: Carbon restoration potential on global land under water resource constraints, Nat. Water, 2, 1071–1081, https://doi.org/10.1038/s44221-024-00323-5, 2024.
  - Pierce, S., Brusa, G., Vagge, I., and Cerabolini, B. E. L.: Allocating CSR plant functional types: the use of leaf economics and size traits to classify woody and herbaceous vascular plants, Funct. Ecol., 27, 1002–1010, https://doi.org/10.1111/1365-2435.12095, 2013.
  - Pokhrel, S. and Sherpa, C.: Analyzing the relationship, distribution of tree species diversity, and above-ground biomass on the Chitwan-Annapurna landscape in Nepal, Int. J. For. Res., 2020, https://doi.org/10.1155/2020/2789753, 2020.
- Poudel, A. S., Shrestha, B. B., Joshi, M. D., Muniappan, R., Adiga, A., Venkatramanan, S., and Jha, P. K.: Predicting the Current and Future Distribution of the Invasive Weed Ageratina adenophora in the Chitwan-Annapurna Landscape, Nepal, Mt. Res. Dev., 40, R61–R71, https://doi.org/10.1659/MRD-JOURNAL-D-19-00069.1, 2020.
- Pugh, T. A. M., Arneth, A., Kautz, M., Poulter, B., and Smith, B.: Important role of forest disturbances in the global biomass turnover and carbon sinks, Nat. Geosci., 12, 730–735, https://doi.org/10.1038/s41561-019-0427-2, 2019.
- Rabin, S. S., Melton, J. R., Lasslop, G., Bachelet, D., Forrest, M., Hantson, S., Kaplan, J. O., Li,
   F., Mangeon, S., Ward, D. S., Yue, C., Arora, V. K., Hickler, T., Kloster, S., Knorr, W.,
   Nieradzik, L., Spessa, A., Folberth, G. A., Sheehan, T., Voulgarakis, A., Kelley, D. I.,
   Colin Prentice, I., Sitch, S., Harrison, S., and Arneth, A.: The Fire Modeling

- Intercomparison Project (FireMIP), phase 1: Experimental and analytical protocols with detailed model descriptions, Geosci. Model Dev., 10, 1175–1197, https://doi.org/10.5194/gmd-10-1175-2017, 2017.
- Satyakam, Zinta, G., Singh, R. K., and Kumar, R.: Cold adaptation strategies in plants—An
   emerging role of epigenetics and antifreeze proteins to engineer cold resilient
   plants, Front. Genet., 13, 909007, https://doi.org/10.3389/fgene.2022.909007,
   2022.
- Sauter, F., Albrecht, H., Kollmann, J., and Lang, M.: Competition components along
   productivity gradients revisiting a classic dispute in ecology, Oikos, 130, 1326–
   1334, https://doi.org/10.1111/oik.07706, 2021.
- Scherstjanoi, M., Kaplan, J. O., and Lischke, H.: Application of a computationally efficient
   method to approximate gap model results with a probabilistic approach, Geosci.
   Model Dev., 7, 1543–1571, https://doi.org/10.5194/gmd-7-1543-2014, 2014.
- Sekar, K. C., Thapliyal, N., Pandey, A., Joshi, B., Mukherjee, S., Bhojak, P., Bisht, M., Bhatt,
   D., Singh, S., and Bahukhandi, A.: Plant species diversity and density patterns
   along altitude gradient covering high-altitude alpine regions of west Himalaya,
   India, Geol. Ecol. Landsc., 8, 559–573,
   https://doi.org/10.1080/24749508.2022.2163606, 2024.
- Shah, S., Shrestha, K. K., and Scheidegger, C.: Variation in Plant Functional Traits along
   Altitudinal Gradient and Land Use Types in Sagarmatha National Park and Buffer
   Zone, Nepal, Am. J. Plant Sci., 10, 595–614,
   https://doi.org/10.4236/ajps.2019.104043, 2019.
- Shrestha, K. B., Hofgaard, A., and Vandvik, V.: Recent treeline dynamics are similar 930 between dry and mesic areas of Nepal, central Himalaya, J. Plant Ecol., 8, 347– 931 358, https://doi.org/10.1093/jpe/rtu035, 2015.
- Sigdel, S. R., Liang, E., Rokaya, M. B., Rai, S., Dyola, N., Sun, J., Zhang, L., Zhu, H., Chettri, N.,
   Chaudhary, R. P., Camarero, J. J., and Peñuelas, J.: Functional traits of a plant
   species fingerprint ecosystem productivity along broad elevational gradients in
   the Himalayas, Funct. Ecol., 383–394, https://doi.org/10.1111/1365 2435.14226, 2022.
- Silva, N., Coelho, A. J. P., and Meira-Neto, J. A. A.: Functional traits patterns along an 938 altitudinal gradient in a large tropical forest region, Flora, 308, 152403, 939 https://doi.org/10.1016/j.flora.2023.152403, 2023.
- Sitch, S., Smith, B., Prentice, I. C., Arneth, A., Bondeau, A., Cramer, W., Kaplan, J. O., Levis,
   S., Lucht, W., Sykes, M. T., Thonicke, K., and Venevsky, S.: Evaluation of ecosystem
   dynamics, plant geography and terrestrial carbon cycling in the LPJ dynamic
   global vegetation model, Glob. Change Biol., 9, 161–185,
   https://doi.org/10.1046/j.1365-2486.2003.00569.x, 2003.
- Smith, B. and Wilson, J. B.: A Consumer's Guide to Evenness Indices, Oikos, 76, 70,
   https://doi.org/10.2307/3545749, 1996.
- Smith, B., Prentice, I. C., and Sykes, M. T.: Representation of vegetation dynamics in the 948 modelling of terrestrial ecosystems: Comparing two contrasting approaches 949 within European climate space, Glob. Ecol. Biogeogr., 10, 621–637, 950 https://doi.org/10.1046/j.1466-822X.2001.00256.x, 2001.
- Smith, B., Wärlind, D., Arneth, A., Hickler, T., Leadley, P., Siltberg, J., and Zaehle, S.:
   Implications of incorporating N cycling and N limitations on primary production
   in an individual-based dynamic vegetation model, Biogeosciences, 11, 2027–
   2054, https://doi.org/10.5194/bg-11-2027-2014, 2014.

https://doi.org/10.5194/egusphere-2025-4821 Preprint. Discussion started: 20 November 2025 © Author(s) 2025. CC BY 4.0 License.

| 955 | Thakur, D. and Chawla, A.: Functional diversity along elevational gradients in the high              |
|-----|------------------------------------------------------------------------------------------------------|
| 956 | altitude vegetation of the western Himalaya, Biodivers. Conserv., 28, 1977-1996,                     |
| 957 | https://doi.org/10.1007/s10531-019-01728-5, 2019.                                                    |
| 958 | Thakur, G., Kumar, P., Bhardwaj, D. R., Prakash, P., and Poonam: Dynamics of                         |
| 959 | aboveground vegetation biomass and carbon stocks along the altitudinal                               |
| 960 | gradients and overstorey composition types in the temperate Himalayan region,                        |
| 961 | Trees For. People, 16, 100553, https://doi.org/10.1016/j.tfp.2024.100553, 2024.                      |
| 962 | Thakur, R. B. and Phulara, N. K.: A Compendium of Tree Species of Nepal, Sarvottam                   |
| 963 | Offset Printing Press (P.) Ltd., 2014.                                                               |
| 964 | Tito, R., Vasconcelos, H. L., and Feeley, K. J.: Mountain Ecosystems as Natural                      |
| 965 | Laboratories for Climate Change Experiments, Front. For. Glob. Change, 3, 38,                        |
| 966 | https://doi.org/10.3389/ffgc.2020.00038, 2020.                                                       |
| 967 | Whittaker, R. H.: Dominance and Diversity in Land Plant Communities: Numerical                       |
| 968 | relations of species express the importance of competition in community                              |
| 969 | function and evolution., Science, 147, 250–260,                                                      |
| 970 | https://doi.org/10.1126/science.147.3655.250, 1965.                                                  |
| 971 | Zhu, L., Zhang, Y., Ye, H., Li, Y., Hu, W., Du, J., and Zhao, P.: Variations in leaf and stem traits |
| 972 | across two elevations in subtropical forests, Funct. Plant Biol., 49, 319-332,                       |
| 973 | https://doi.org/10.1071/FP21220, 2022.                                                               |
|     |                                                                                                      |