# Peer review of "Vegetation Patterns and Competitive Dynamics along Elevation Gradients: Interactions between Environmental Factors and Vegetation in the Central Himalayas"

_EGUsphere, 2025_

## Author Comment (AC1)

Response to RC1

Reviewer comments are in *Italics*. Author responses are in normal type (red).

*This manuscript aims to investigate roles of abiotic stress and competitive interactions in shaping plant community assembly by using dynamic vegetation model with plant functional types. The approach seems interesting however, I found several methodological uncertainties, potential flaws, styles and errors throughout the manuscript that should be addressed to make it convincing.*

We thank the reviewer for this encouraging assessment. We agree that the manuscript would benefit from targeted clarification and refinement, and we have outlined below the specific revisions to implement in response to each point.

*Introduction: The foundation of the story is week for me. The knowledge gap or research questions and hypothesis are missing. Introduction is a bit general for me. Wise to make it more appealing by synthesizing key ecological aspects. Basically, it needs to present the significance of this study. For instance, what are the knowledge gaps based on previous studies in the Himalayan regions. How this study helps to advance our understanding on plant community assembly. There are several trait-based (morphological to elemental scale), field observation-based studies along the elevation gradients or focused on particular ecotones, which are overlooked in the manuscript. Those studies might be helpful to improve it.*

The goal of this paper is to evaluate how abiotic stress and competitive interactions jointly shape plant community assembly, structure, and productivity along the elevation gradient in the Central Himalayas, using a dynamic vegetation model parameterized with empirical trait and life-history data. Specifically, we aim to test the hypothesis of a vegetation composition and productivity transition from competition-driven (realised niche) dynamics at lower elevations to stress-driven (physiological niche) dynamics at higher elevations. We agree that the Introduction section does not currently communicate these objectives, research questions, and hypotheses with sufficient clarity or synthesis. We will revisit and restructure the Introduction to explicitly state the study's goals, identify key knowledge gaps in Himalayan plant community assembly, synthesize relevant trait-based and field observation studies, and clearly articulate how this modelling framework advances understanding beyond existing empirical and regional-scale studies.

*The methodology needs several clarifications. There are several issues on data (parameters) used in the model simulation (see specific comments as well) which weakens the robustness of the model. For instance, Avolio et al. (2019) developed equation based on grassland plot data including species removal approach, it is not clear how authors link it with carbon masses. It is not clear, how this approach linked with observed traits data by*

*Maharjan et al. 2023? Similarly, as Maharjan et al. (2021) collected the traits using common tree species, several deciduous species did not sample as they lost their leaves at the time of fieldwork (please see Maharjan et al. 2021). I am very curious, is these data really applicable to know the local species richness and evenness?*

We appreciate the reviewer's concern regarding the robustness of the parameterisation and the linkage between empirical trait data, competition metrics, and simulated carbon dynamics. In this study, we adopt a modified formulation based on Avolio et al. (2019) to quantify PFT performance under full competition and reduced-competition conditions (removing all tree PFTs except the targeted one from the system). While Avolio et al. (2019) derived this relationship from grassland species-removal experiments, we use the conceptual framework, modified rather than the original empirical coefficients. We will clarify this in the method section.

The empirical trait data collected by Maharjan et al. (2021) represent 31 dominant tree species that account for the majority of above-ground carbon production and structural biomass along the Himalayan elevation gradient. These species were used to derive 13 functionally distinct PFTs through a hierarchical clustering approach, capturing major plant strategies and trait variability relevant to competition, stress tolerance, and growth form. Although some deciduous species were not sampled during leaf-off periods, we have 7 deciduous PFTs, and the parameterisation focuses on structural, allometric, and life-history traits that are expected to be representative of all deciduous species found along the gradient. We will elaborate on how these PFTs represent regional functional traits and plant strategies in the method and supplementary sections.

We emphasize that the objective of the PFT framework is not to reproduce local species richness or plot-level evenness, but to represent dominant functional strategies that control ecosystem structure and carbon dynamics at landscape-to-regional scales. We acknowledge that we calculated PFTs level evenness and richness using fractional coverage, which does not reflect local species evenness and richness. We will clarify this conceptual distinction in the Methods and Discussion sections, explicitly stating the scope and limitations of trait-based PFT representation and its implications for interpreting simulated patterns of diversity and evenness.

*It was not mentioned if the occurrence pixels. For example, sometimes the pixels be in the forest, doesn't fall to the forest but an open area, which is clearly an erroneous or inaccurate observation. The data point checking and cleaning could be done to ensure their correctness*

The model simulations represent ecosystem structure and functioning across the full climatic gradient, independent of whether a given grid cell or simulated patch is currently forested or non-forested. In contrast, the observational above-ground biomass data used for evaluation (Khanal and Boer, 2023) are restricted to forested areas. Accordingly, no additional filtering of simulated pixels was applied to match land-use categories. The

objective of this comparison is therefore to assess whether simulated and observed values exhibit consistent patterns across the elevation gradient, rather than to perform pixel-level validation. We will clarify this modelling rationale and explicitly acknowledge the resulting scale and conceptual mismatch between simulated potential vegetation and land-use–dependent observational data in the Methods and Discussion sections.

*Result section is too comprehensive and very difficult to figure out the key findings. Better to present figures with key results in the main text and other can be transferred to the supporting information. I would suggest just to present the result. Don't mix it with some explanations.*

We will revisit the results section and, if possible, restructure it. We will move any explanatory interpretation to the discussion and focus solely on presenting results in the present section.

*The overall discussion is not well written for me. Mostly, authors just present their results and compare them with other similar studies and fail to provide scientific evidence or possible ecological mechanisms to support their results. Thus, it needs to synthesize the results rather than presenting results directly. Also, key results should be highlighted and justified with scientific evidences. Deeper discussion needed including the mechanisms why and how you obtained such results? What are the implications of these results under climatic changes? How PFTs drive ecological niche formation, how traits explain it, what are the ecological mechanisms and what are their ecological implications. It warrants deeper discussion and wide literature review.*

We will substantially revise the Discussion section focusing on underlying ecological mechanisms (e.g. abiotic filtering, competition asymmetry, trait-mediated niche differentiation), and discuss broader implications under different growth conditions. We will explicitly address how PFTs and traits drive niche formation and community assembly in our model along elevation gradients, supported by a more targeted literature review.

*L130: please cross-check the sentence. Generally, precipitation peaks above 1500-2500 m and it decreases from around 3000 m.*

We will cross-check the data and reference.

*L198: How do you define the wet days or dry days? What are the criteria?*

The Number of wet days is defined as a day with non-zero precipitation(>0mm) and annual counts are obtained for model simulation. We will edit the text accordingly in the Methods section.

*L215-220: how do you define these functional groups? What are the criteria? It should be well described and methods should be reproducible.*

We note that in the Methods section, PFTs are defined using a divisive hierarchical clustering approach. We agree that the current description could benefit from further elaboration. We will expand the Methods section to clearly describe the criteria and variables used in the clustering to ensure reproducibility. Detailed information on the clustering procedure, trait selection, and resulting PFT definitions and parameters will be provided in the Supplementary Material.

*Table 1: What are the sources of these data? source should be provided.*

We will add the source of data in the table footer.

*As I know authors used tree traits data ranging from about 100 m to 3800 m (see Figure 1 Maharjan et al. 2021), how model is simulated up to around 5800 m? Generally, trees are found up to around 4000 m, do this model is also applicable to simulate different ecosystems (such as alpine grasslands). This may create several issues about the robustness of the model. Since authors highlighted of the significance of study is comparing simulated and observed data (see L112-113), there is large gaps.*

The empirical tree trait data used for parameterisation span elevations from approximately 100 to 3800 m (Maharjan et al., 2021) and were used to define regional tree PFTs and to parameterise the model. Model simulations, however, extend to higher elevations (up to ~5800 m) to capture the full climatic gradient, including the transition from forested ecosystems to alpine and nival zones. Above the upper treeline (≈3800–4200 m in the Himalayas), tree PFTs are progressively excluded by temperature constraints and carbon balance limitations within the model. While occasional occurrences of tree PFTs may still appear at the grid level due to stochastic establishment processes, ecosystem structure at these elevations is overwhelmingly dominated by $C_3$ grass PFTs, with negligible woody biomass. Simulated vegetation above the treeline therefore, represents alpine grasslands or sparsely vegetated systems rather than forest ecosystems, consistent with ecological expectations.

We acknowledge that the field observations used for evaluation (Khanal and Boer, 2023) are restricted to forested areas. Accordingly, the purpose of this comparison is not point-by-point validation across the full elevation range, but to validate the broader patterns in above-ground biomass distribution. We will clarify this explicitly in the Methods and Discussion sections and clearly distinguish between evaluated forested elevations and higher-elevation model outputs, which are intended to illustrate emergent responses to climatic stress rather than to make direct comparisons with plot-level observations.

*Figure 3: above-ground carbon mass or above-ground biomass? some places authors mentioned carbon mass, some places biomass, it makes lots of confusions. The terminologies should be consistent. As I see in Khanal and Boer (2023), it might be above-ground biomass. As Forest lines normally up to around 4000 m and Nepal's forest inventory covers only forested areas, I am surprising to see the some observed data close to 6000 m. It indicates there are some errors using the plot-level data.*

We will standardize terminology throughout the manuscript (e.g. above-ground biomass vs above-ground carbon mass). Elevation values for each observed and simulated grid cell were extracted from the DEM. We will also re-examine the dataset used in Figure 3, clarify its spatial coverage, and explicitly discuss any apparent anomalies or uncertainties related to elevation limits of forest inventories.

*L335: How did you simulated LAI? it is not mentioned in the methodology section.*

In the model, LAI is calculated dynamically from simulated leaf carbon pools using PFT-specific SLA, with LAI directly linked to carbon allocation, phenology, and canopy structure. Leaf carbon allocation is influenced by growth conditions and competition, and LAI emerges from the balance between leaf growth and turnover rather than being prescribed. We will incorporate this description into the Methods section to clearly explain how LAI is simulated and how it interacts with carbon allocation and canopy structure within the model framework.

*L342: How do you defined C4 and C3 grasses, as both types grasses found abundantly along the gradient.*

In this simulation, we used default $C_3$ and $C_4$ grass with default parameters as explained in Peng et al. (2024). In our model, $C_3$ and $C_4$ grasses are represented as distinct PFTs that differ in their photosynthetic pathways and associated physiological traits. $C_4$ grasses are characterized by higher temperature optima for photosynthesis, greater water-use efficiency, and reduced photorespiration, whereas $C_3$ grasses have lower temperature optima and are more competitive under cooler and less water-limited conditions. These differences are reflected in PFT-specific parameterization of photosynthesis, temperature response functions, and phenology. The relative abundance of $C_3$ and $C_4$ grasses along the elevation gradient therefore emerges from the interaction between climate (particularly temperature), competition, and disturbance within the model. We will elaborate on the Method section to briefly discuss $C_3$ and $C_4$ grasses' PFTs.

*Figure 4: For other parameters such as above-ground biomass and bole height, but just showing simulated LAI don't provide any insights whether it works well or not.*

We agree that only simulated LAI is currently shown. Independent PFT-level LAI observations are not available along the elevation gradient, which limits a direct

validation of simulated LAI. As an alternative, we will compare simulated LAI with MODIS LAI products and carefully discuss uncertainties arising from differences in spatial scale and aggregation.

*Figure 5 caption: source of observed data should be acknowledged.*

We will add the source in the figure caption.

*L451: Is there any pine species up to 4500 m? In my understanding, it is totally wrong.*

We agree that *Pinus* species do not extend to ~4500 m in the Himalayas. In our model, three coniferous PFTs represent distinct elevational adaptations: (i) a sub-tropical coniferous PFT representing *Pinus roxburghii*, which dominates approximately between 400–2000 m; (ii) a temperate coniferous PFT representing *Pinus wallichiana*, which dominates approximately between 1800–3600 m; and (iii) an alpine coniferous PFT representing cold-tolerant conifer strategies characteristic of higher elevations. We will revise the manuscript to remove specific references to *Pinus* at higher elevations and instead describe the pattern in terms of evergreen conifer functional types. We will further clarify that conifer signals at higher elevations are more consistent with taxa such as *Abies* and *Juniperus*, and emphasize that the model results reflect functional responses to climatic constraints rather than species-level distributions.

*L467: As authors performed compare the patterns separately, thus, in my understanding, not well evaluated the complex interactions. To evaluate the overall interactions, it is wise to quantify the relative importance of different biotic and abiotic variables on PFTs.*

We acknowledge that we did not explicitly quantify the relative importance of individual biotic and abiotic drivers using statistical attribution methods. Instead, in this study, complex interactions between biotic and abiotic variables are evaluated as emergent phenomena by analyzing PFTs' structure, composition, and productivity along the elevation gradient. Signals of interaction arise from the combined effects of plant competition, trait-based parameterization, and environmental constraints (particularly temperature) operating simultaneously along the elevation gradient, leading to emergent patterns in PFT composition, structure, productivity, and performance in abundance. By analysing PFTs' performance under competitive and stress-dominated conditions, we infer how biotic interactions and abiotic limitations jointly shape structure, composition, and productivity. Formal quantification of relative importance is unfortunately out of scope for this study. We will clarify this distinction in the manuscript and highlight such analyses as a potential avenue for future work in limitation section.

*L521-522: Actually, high elevations, particularly above 3000 m deciduous Betula utilis (Himalayan birch) is one of the most dominant species. Thus, this statement could be problematic.*

We will edit it and elaborate on the discussion section to acknowledge how generalized PFTs can have limitations in representing individual species in limitation sections.

*L633: Not well discussed whether simulated results synchronized with observed data or not and reasons behind these.*

We will revise it.